# Drought, armed conflict and population mortality in Somalia, 2014–2018: A statistical analysis

**Abdihamid Warsame, Séverine Frison, Francesco Checchi** *

Faculty of Epidemiology and Population Health, Department of Infectious Disease Epidemiology, London School of Hygiene and Tropical Medicine, London, United Kingdom

* francesco.checchi@lshtm.ac.uk

## Abstract

During 2010–2012, extreme food insecurity and famine in Somalia were estimated to account for 256,000 deaths. Since 2014 Somalia has experienced recurrent below-average rainfall, with consecutive failed rains in late 2016 and 2017 leading to large-scale drought, displacement and epidemics. We wished to estimate mortality across Somalia from 2014 to 2018, and measure the excess death toll attributable to the 2017–2018 drought-triggered crisis. We used a statistical approach akin to small-area estimation, and relying solely on existing data. We identified and re-analysed 91 household surveys conducted at the district level and estimating the crude (CDR) and under 5 years death rate (U5DR) over retrospective periods of 3–4 months. We captured datasets of candidate predictors of mortality with availability by district and month. We also reconstructed population denominators by district-month combining alternative census estimates and displacement data. We combined these data inputs into predictive models to estimate CDR and U5DR and combined the predictions with population estimates to project death tolls. Excess mortality was estimated by constructing counterfactual no-crisis scenarios. Between 2013 and 2018, Somalia's population increased from 12.0 to 13.5 million, and internally displaced people or returnees reached 20% of the population. We estimated an excess death toll of 44,700 in the most likely counterfactual scenario, and as high as 163,800 in a pessimistic scenario. By contrast to 2010–2012, excess deaths were widespread across Somalia, including central and northern regions. This analysis suggests that the 2017–2018 crisis had a lower, albeit still very substantial, mortality impact than its 2010–2012 predecessor. Despite modest elevations in death rate, crisis conditions were widespread and affected a population of millions. Humanitarian response to drought-related crises in Somalia needs to be strengthened, target the most vulnerable and emphasise very early interventions.

## Background

Somalia has experienced recurrent climate- and armed conflict-driven crises over the past 30 years. As a result, the population has limited resilience and is vulnerable to shocks brought on by such crises. In 2010–2012, largely as a result of consecutive failed rains and limited

**Data Availability Statement:** All data and analysis scripts, along with instructions for replicating the analysis, are available on https://github.com/

franceschechi/mortality_small_area_
estimation.

**Funding:** This work was funded by the UK
Department for International Development, now
part of the UK Foreign Commonwealth and
Development Office, FCDO (to AW, SF and FC)
through the Research for Evidence Division (RED).
However, the views expressed and information
contained in it are not necessarily those of or
endorsed by FCDO, which can accept no
responsibility for such views or information or for
any reliance placed on them. Additional funding for
all authors (AW, SF, FC) came from UK Research
and Innovation as part of the Global Challenges
Research Fund, grant number ES/P010873/1. The
funders had no role in study design, data collection
and analysis, decision to publish, or preparation of
the manuscript.

**Competing interests:** The authors have declared
that no competing interests exist.

humanitarian access and assistance, a famine occurred across south-central Somalia. A study
commissioned by the United Nations (and co-authored by some of us) estimated 256,000
deaths attributable to exceptional food insecurity during this period, of which approximately
half in children under 5 years [1].

The two years following the famine saw modest improvements, coinciding with intensified
humanitarian assistance and moderate rainfall. By end 2014, however, reduced humanitarian
access due to renewed insecurity, as well as below-average rainfall, had returned pockets of
Somalia to acute emergency conditions [2]. The years 2015 and 2016 saw a mixture of failed
seasonal rains and exceptional flooding, complicated by the return of Somalis from Kenya ref-
ugee camps and war-stricken Yemen [3].

In 2016, the sustained underperformance of seasonal rains in Puntland and Somaliland
triggered both governments to declare states of emergency. Most regions of Somalia experi-
enced far below-average rainfall in September-December 2016, compromising key harvests
and livestock viability [4]; as shown in Fig 1, in some regions drought persisted into 2017, and
large movements of internally displaced persons (IDPs) were seen, topping 2.6 million by end
2018. Large-scale epidemics of cholera and measles, typical of food insecurity and displace-
ment situations, also occurred during this period [5]. Abundant rainfall was however recorded
from mid-2018. We aimed to update our 2010–2012 analysis by estimating mortality across
Somalia during 2014–2018, while also quantifying the excess death toll attributable to drought
during 2017–2018.

## Methods

### Study population, period and outcomes

Our analysis covered all of Somalia (including Somaliland and Puntland) and was stratified by
district and month, spanning the period January 2014 to December 2018, with excess mortality
estimated for 2017–2018, the drought-affected period of interest. We sought to estimate all-age
and under 5y mortality: corresponding indicators are the crude death rate (CDR) and the
under 5 years death rate (U5DR); the latter, unlike under 5y mortality ratios (which measure
the probability of survival to age 5y), expresses the incidence of death among children.

### Study design

Our method consists of six mostly sequential steps, summarised in Fig 2, and is broadly classi-
fiable as a small-area estimation approach. It rests on constructing a statistical model to predict
mortality based on a combination of previously collected data, and using the model, in con-
junction with population denominators, to retrospectively estimate death rates and tolls across
the crisis-affected population and period of interest. The model is also used to predict mortal-
ity in counterfactual scenarios of no crisis, and the difference between these counterfactual sce-
narios and the estimated mortality is taken as the excess, crisis-attributable mortality. Details
on theory, data management, analytic steps and R software [6] implementation are presented
in Checchi et al. [7] and S1 Text. All data and analysis scripts are uploaded to https://github.
com/franceschechi/mortality_small_area_estimation. Below, we summarise data sources,
analysis steps and specify adaptations made for the Somalia context.

### Data sources and management

All data were previously collected for routine humanitarian response and/or health service
provision purposes, and were either in the public domain or shared in fully anonymised
format.

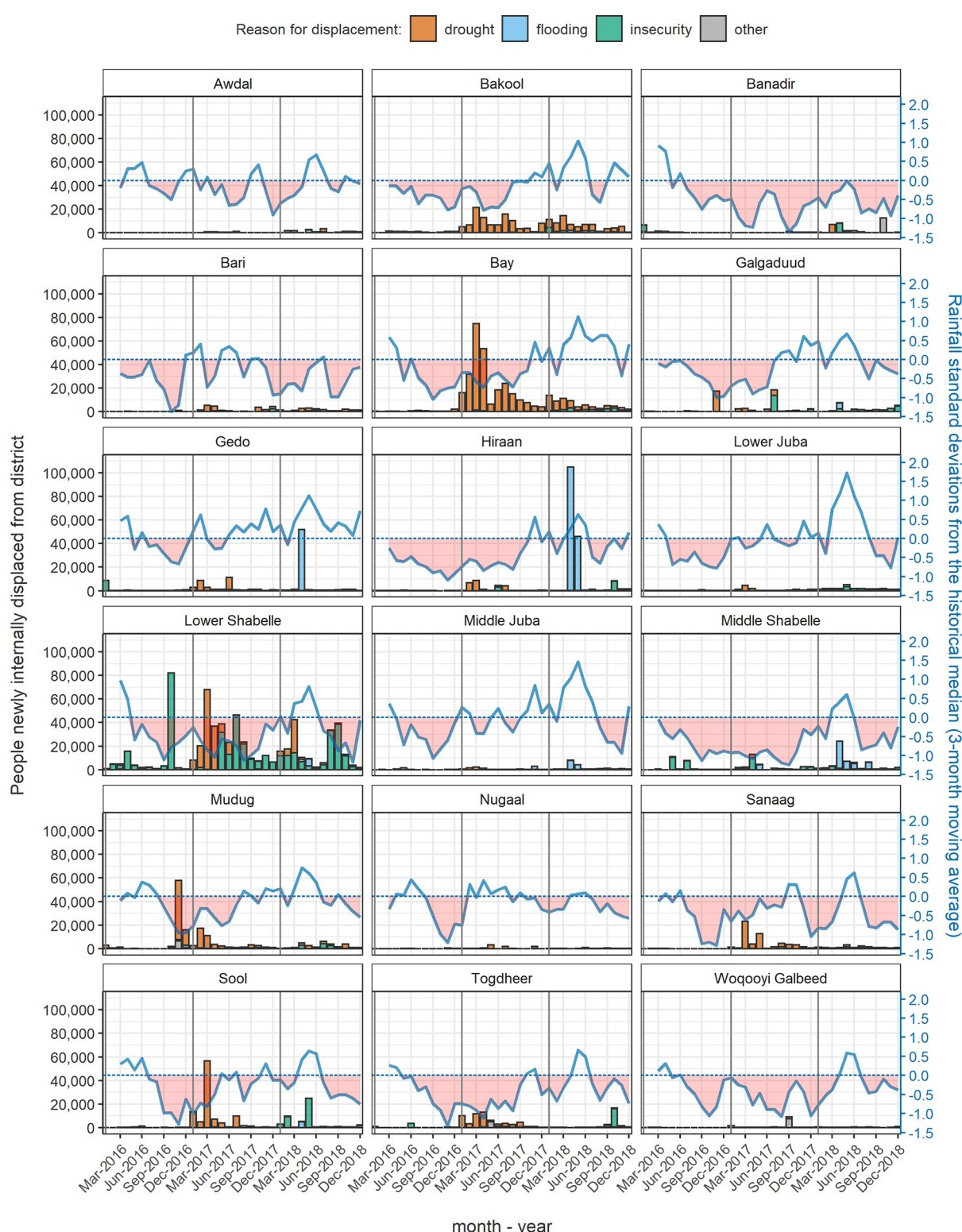

**Fig 1. Rainfall levels compared to the historical average (dotted line) and number of people newly displaced (bars), by region of Somalia, 2016–2018.** See Methods for data sources.

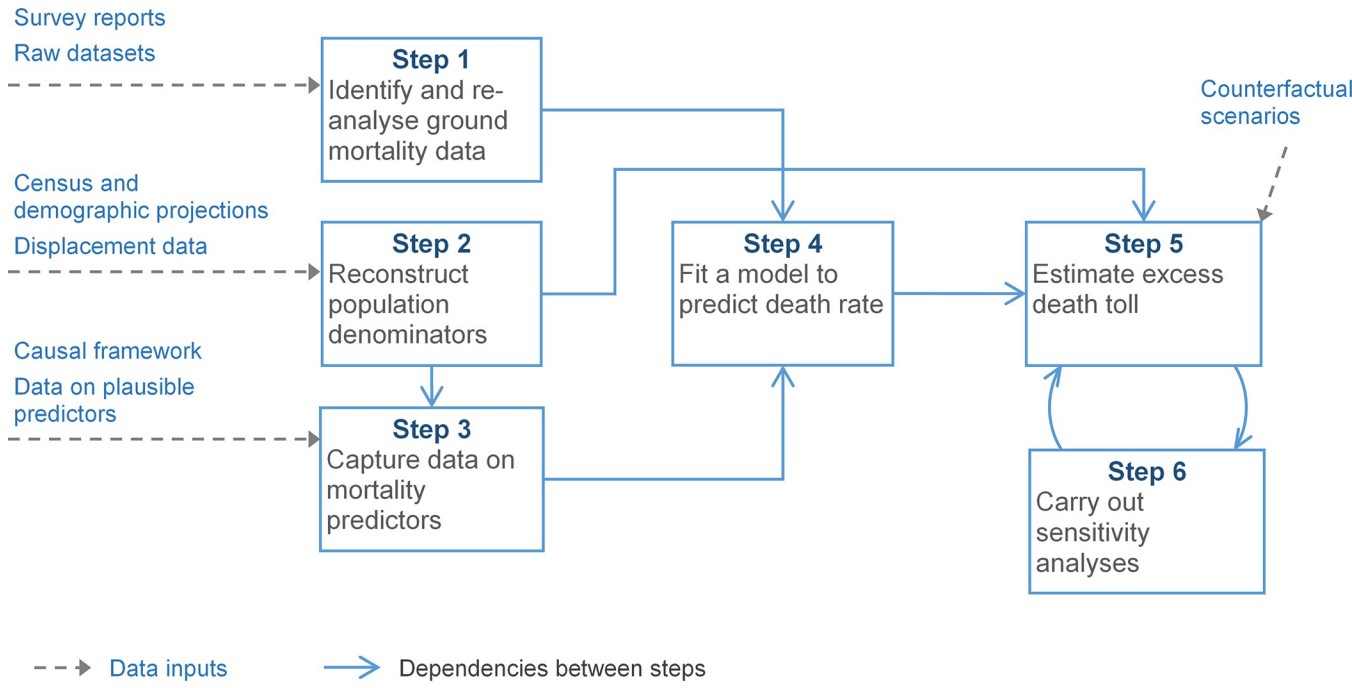

**Fig 2. Schematic of estimation steps and required data inputs.**

**Ground mortality data.** We sourced all available retrospective household surveys conducted during the analysis period by different humanitarian actors using the Standardised Monitoring of Relief and Transitions (SMART) methodology [8]: while these surveys are primarily conducted to estimate the prevalence of acute malnutrition, they often include a mortality questionnaire module, which elicits information from respondents on the demographic evolution (composition, births, deaths, in- and out-migration) of their households during a 'recall' period of 3–4 months [9,10]. Prior to 2016, SMART surveys often relied on an 'aggregate' questionnaire, which elicits only numbers of household members and demographic events; since 2016, only a more detailed 'individual' questionnaire has been used, whereby each household member present during the recall period is listed. Surveys use systematic random sampling or two-stage cluster sampling with probability of cluster selection proportional to size; we used surveys whose sampling universe was an entire district, or urban areas or IDP settlements within a district. We extracted meta-data on, cleaned and re-analysed all surveys with datasets available to us, estimating CDR, U5DR and other demographic indicators using generalised linear models at the household level, assuming a Poisson distribution offset by household person-time at risk, and adjusting standard errors for intra-cluster correlation. We also computed a quality score for each survey based on SMART criteria for anthropometry [7].

We had access to 201 surveys that estimated retrospective mortality in Somalia between 2013 and 2018 inclusive (Fig 3). Most (175, 87.1%) were conducted by the FSNAU, and the remainder by four other agencies. A raw dataset was available for all surveys, but a detailed report only for 28 (13.9%). We excluded 104 (51.7%) of the surveys from analysis. The main reason for exclusion (n = 87) was because surveys were designed to be representative of livelihood zones rather than districts: as population and predictor datasets were mostly stratified by district, analysing these surveys would have required additional assumptions (livelihood zones straddle multiple districts, and vice versa). Other exclusion reasons were unclear sampling frame (n = 11) and non-standard questionnaire and unclear recall period (n = 6). This left 97

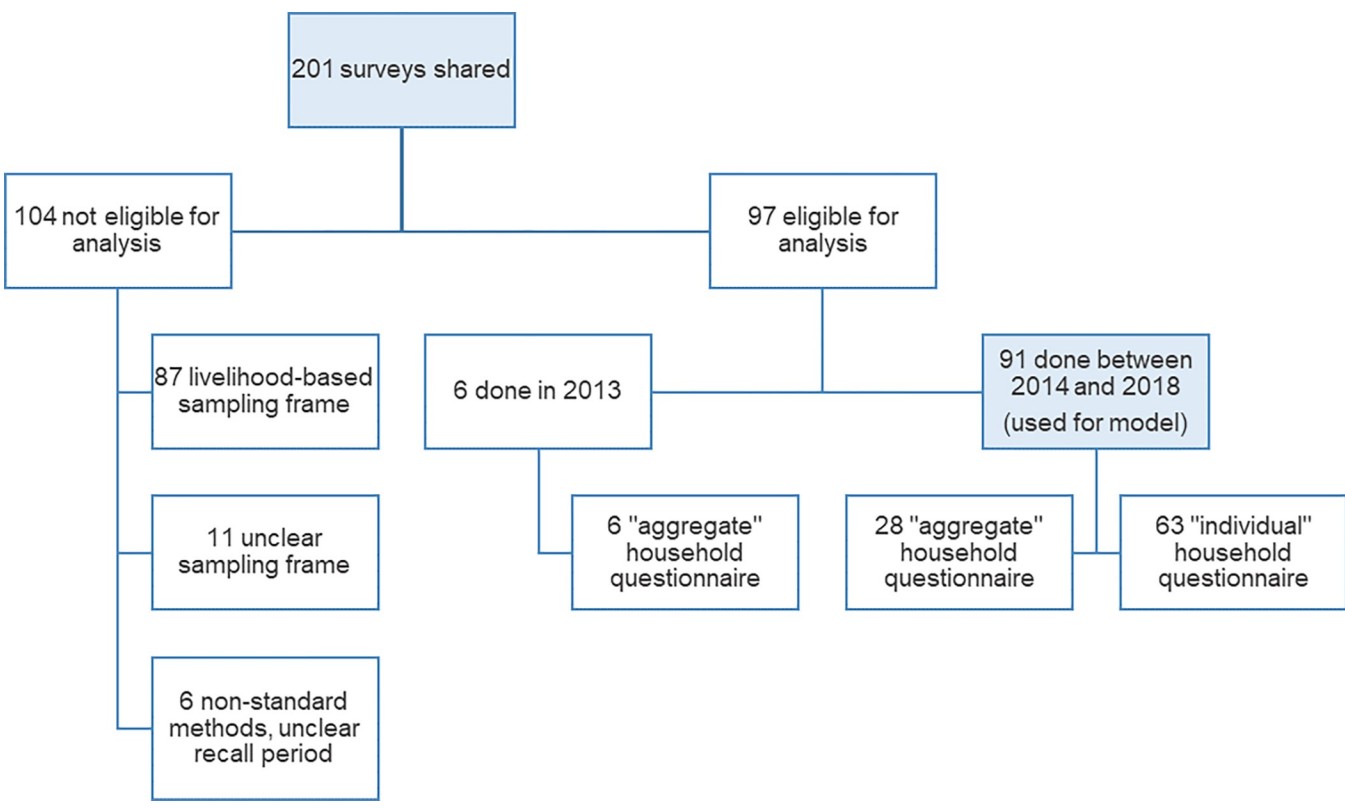

**Fig 3. Schematic of mortality survey availability.**

eligible surveys, of which six done in 2013 were not used for modelling as predictor data coverage was insufficient during this year. Of the 91 surveys entered into statistical models, most (n = 63) used an individual questionnaire; the remainder (n = 28) used the older aggregate questionnaire. Survey availability by district-month is shown in Fig A in S1 Text.

**Demographic and displacement data.** Somalia last conducted a census in 1975. The four alternative census estimates known to us, and falling within the analysis period, consisted of (i) the March 2014 United Nations Population Estimation Sample Survey (UNPESS), carried out by the United Nations Population Fund, based on a stratified sample of clusters within urban, rural, IDP and nomadic communities [11]; (ii) January 2015 estimates from AfriPop (now WorldPop: https://www.worldpop.org/), which used a validated statistical model to estimate population density by 100m$^2$ pixel based on remote sensing [12] (iii) December 2018 data from the Polio Eradication Initiative, which in Somalia updates target vaccination population denominators through active enumeration (non-public); and (iv) January 2019 Expanded Programme on Immunization data, adjusted based on vaccination campaign performance data (non-public).

Internal displacement data came from the United Nations High Commissioner for Refugees (UNHCR)-led Protection and Return Monitoring Network (PRMN) [13], which tracks displacement and returnee movements from one district to the next as reported by a network of humanitarian and civil society actors, with monthly data available from January 2016 to December 2018. Data also include the reported main reason for displacement. We also reviewed situation reports on the humanitarian ReliefWeb information platform (www.reliefweb.int) and on the UNHCR web site (www.unhcr.org) to approximate refugee flows in and out of districts.

**Data on predictors of mortality.** Referring to a published causal framework [14] of excess mortality in crises, we searched for online open-access data sources and held discussions with likely data holders (United Nations agencies, humanitarian coordination mechanisms, Somali government agencies) to identify and source possible predictors of death rate for which data-sets with reasonable completeness ($\geq 70\%$ complete for $\geq 70\%$ district-months) were available. Candidate predictors for which we found data are detailed in Table 1. Where relevant, we divided predictor values by reconstructed population denominators (see below). Market price data, necessary to compute terms of trade indicators, featured substantial missingness, which we dealt with through manual imputation, by attributing to district-months with missing data a weighted mean of values from districts in the same region (relative weight of 1) and values from districts outside the region (relative weight 0.3); weights were chosen arbitrarily. After imputation, we also smoothed.the price time series using the smooth.spline R package, with spar (smoothing function) value set to 0.3, selected after extensive visual inspection.

**Reconstruction of population denominators.** The UNPESS demographic survey esti-mated the prevalent proportion of IDPs within each district as of start 2014. As no incident dis-placement data were available before 2016, we simply assumed no further displacement during 2014–2015: this assumption may be reasonably accurate, given that throughout Somalia this period saw no major acute emergencies and relatively low levels of insecurity; humanitarian actors we liaised with also suggested that this was a plausible assumption. From 2016 onwards, we computed a monthly net flow of IDPs and refugees in and out of each district, based on UNHCR data. We applied these flows, along with an assumed 2.1% yearly growth rate (differ-ence between UN-projected crude birth and death rates [21]) to the four alternative census esti-mates, back- or forward-calculating monthly district populations from the timepoint at which each estimated was centred. We consulted documentation available for each of the four demo-graphic estimates to score their quality on a scale from 0 to 1, based on published criteria [22]. We used these scores as a weight to average the alternative population time series into a single estimate for our analysis. Lastly, we computed under 5y populations by applying the median proportion of children in this age group (25.0%) across the household surveys we reviewed.

## Predictive models

We evaluated the accuracy of different candidate sets of predictors to predict mortality (CDR and U5DR) by fitting quasi-Poisson models of these predictors to mortality survey data (household-level counts of deaths, offset by household person-time at risk during the recall period); average values of each predictor over the survey's recall period, and in the district surveyed, were used. After screening out predictors with poor fit, we did a brute-force search across all possible candi-date models, selecting the final model out of the top 20% best-fitting alternatives. Model selection procedures are detailed in Checchi et al. [7]. Briefly, the validity (predictive accuracy) of each can-didate model was assessed primarily through ten-fold cross-validation within the full dataset, with the mean Dawid–Sebastiani score across all folds as the metric of fit (a low score indicates good fit). As a secondary validation, we also fit models on a 80% random 'training' sample of sur-veys and observed predictive accuracy on the remaining 20% 'holdout' sample. Among shortlisted models, we manually selected one that had the highest possible predictive accuracy on both cross-validation and holdout data while also being as parsimonious as possible and featuring predictors (e.g. epidemic occurrence) that were plausibly sensitive to crisis conditions, and thus could be used to meaningfully construct counterfactual scenarios. We explored both categorical and con-tinuous versions of variables, and lags up to 6 months where plausible. We introduced random effects (survey cluster and district) and plausible interactions, but retained neither as they wors-ened model fit. Lastly, we computed robust coefficient standard errors.

**Table 1. Candidate predictors of mortality considered in the analysis.**

| Predictor | Variable(s) | Domain | Time span of availability | Source(s) (citation and URL if public) | Notes and assumptions |
|---|---|---|---|---|---|
| Administrative level | Administrative entity within Somalia | (various) | n/a (static variable) | n/a | Somaliland, Puntland, south-central Somalia |
| Rainfall | Total rainfall (mm) | Climate | 2013 to 2018 | Climate Engine [15,16] | |
| | Mean of Standard Precipitation Index | | 2016 to 2018 | | Compares current rainfall with historical averages. |
| Vegetation density | Normalised Difference Vegetation Index | Climate | 2013 to 2018 | Food Security and Nutrition Analysis Unit—Somalia (FSNAU) [17] | |
| Incidence of armed conflict events† | events per 100,000 population deaths per 100,000 population | Exposure to armed conflict / insecurity | 2010 to 2018 | Armed Conflict Location & Event Data Project (ACLED) [18]: https://www.acleddata.com/ | Meta-data on individual armed conflict events based on extensive review of multi-language media sources and other public information. |
| Incidence of attacks against aid workers† | deaths per 100,000 population injuries per 100,000 population | Exposure to armed conflict / insecurity | 2010 to 2018 | Aid Worker Security Database [19]: https://aidworkersecurity.org/incidents | Data on various types of attacks to aid workers, capturing information from media sources, aid organisations and security actors. |
| Proportion of IDPs | proportion of IDPs among total district population | Forced displacement | 2016 to 2018 | As estimated by authors. | |
| Main local livelihood type | Pastoral, agropastoral, riverine and urban | Food security and livelihoods | n/a (static variable) | FSNAU [20] | Assumed to be constant over time. |
| Water price | Price of 200L drum of water in Somali Shillings | Food insecurity and livelihoods | 2013 to 2018 | FSNAU [17] | |
| Terms of trade purchasing power index | Kcal equivalent of local cereals that an average local-quality goat can be exchanged for | Food insecurity and livelihoods | 2013 to 2018 | Calculated by the authors based on FSNAU price data [17] from 100 sentinel markets. | See S1 Text. |
| | Kcal equivalent of local cereals that can be purchased with an average daily labourer wage | | | | |
| Malnutrition incidence† | cases of severe acute malnutrition admitted to treatment programmes per 100,000 population | Nutritional status | 2011 to 2018 | Nutrition Cluster, Somalia | Non-publicdata. |
| | cases of global acute malnutrition admitted to treatment programmes per 100,000 population | | 2013 to 2018 | | |
| Cholera incidence† | cases per 100,000 population | Disease burden (epidemic) | 2013 to 2018 | FSNAU [17] | Suspected and confirmed cases. No cases reported before 2015. |
| Measles incidence† | cases per 100,000 population | Disease burden (epidemic) | 2013 to 2018 | FSNAU [17] | Suspected and confirmed cases. No cases reported before 2015. |
| Malaria incidence† | cases per 100,000 population | Disease burden (epidemic) | 2013 to 2018 | FSNAU [17] | Suspected and confirmed cases. No cases reported before 2015. |
| Humanitarian actor presence† | Ongoing humanitarian projects per 100,000 population (all sectors) | Humanitarian (public health) service functionality | 2010 to 2018 | United Nations Office for Coordination of Humanitarian Affairs, Somalia | Proxy of intensity of humanitarian response. Non-public data. |
| | Ongoing projects per 100,000 population (health, nutrition and water, hygiene and sanitation) | | | | |
| Measles vaccination | Coverage of one dose of measles-containing vaccine among children | Humanitarian (public health) service coverage | 2017 to 2018 | World Health Organization, Somalia | Based on programmatic data (administrative coverage). Age range unclear. Non-public data. |

(*Continued*)

**Table 1.** (Continued)

| Predictor | Variable(s) | Domain | Time span of availability | Source(s) (citation and URL if public) | Notes and assumptions |
|---|---|---|---|---|---|
| Food security humanitarian services† | Proportion of the population that are a beneficiary of any food security service | Humanitarian (public health) service coverage | Jan 2013 to Apr 2018 | Food Security Cluster, Somalia | Non-public data. |
| | Proportion of the population that are a beneficiary of cash-based food security services | Humanitarian (public health) service coverage | | | |
| | Proportion of the population that are a beneficiary of food distributions | Humanitarian (public health) service coverage | | | |
| Quality of SAM treatment | Proportion of SAM admissions that exit the treatment programme cured | Humanitarian (public health) service quality | 2011 to 2018 | Nutrition Cluster, Somalia | Non-public data. |

† Divided by district population estimates to obtain a population rate.

## Estimation of excess death tolls

Working with the selected models for CDR and U5DR, we predicted death rates and (after multiplying by population denominators) tolls, by district-month. We also estimated what mortality would have been in the most likely, reasonable best- and reasonable worst-case counterfactual scenarios (Table 2): each scenario was constructed by varying both population denominators and model predictor values in accordance with assumptions on what would have happened in the absence of a crisis. We generated 10,000 bootstrap sets of actual and counterfactual predictions by sampling from model error distributions, and for each set computed excess mortality as the difference. We then computed point estimates (modes) and 95% percentile intervals from the distributions of bootstrap samples.

## Sensitivity analyses

We wished to examine the sensitivity of estimates to two key possible error sources. Firstly, we explored potential systematic or unsystematic error in the population and displacement input

**Table 2. Most likely, reasonable worst-case and best-case counterfactual scenarios.**

| Variable | Most likely scenario | Worst-case scenario | Best-case scenario |
|---|---|---|---|
| Predictors | | | |
| SAM admissions rate | Month-specific median of 2014–2016 values within each stratum | Month-specific 75th percentile of 2014–2016 values within each stratum | Month-specific 25th percentile of 2014–2016 values within each stratum |
| Measles incidence rate | Month-specific median of 2014–2016 values within each stratum | Same as in actuality | No measles cases |
| Malaria incidence rate | Month-specific median of 2014–2016 values within each stratum | 25% of actuality† | Same as in actuality† |
| Armed conflict event rate | Median of 2014–2016 values within each stratum | Same as in actuality | 25% of actuality |
| Population | | | |
| Internal displacement flows | No displacement due to drought | Same as in actuality | No new displacement |
| Refugee flows | Same as in actuality | Same as in actuality | Same as in actuality |

† As higher malaria incidence predicted lower mortality, we assumed a low incidence of malaria for the worst-case scenario, and vice versa. See Discussion.

data by replicating the analysis for different combinations of relative errors in district population and displacement estimates. Secondly, we considered possible under-reporting of under 5y deaths, as noted in a previous South Sudan analysis [23], and suggested by the surprisingly low ratio of U5DR to CDR (see below). For different assumed proportions of child deaths not reported, we computed the corresponding 'unseen' number of deaths in each survey; then, for each of 10,000 bootstrap replicates, we attributed these unseen deaths at random across the surveyed households and ran all subsequent steps of the analysis.

## Ethics

The study was approved by the Ethics Committee of the London School of Hygiene & Tropical Medicine (ref. 15334) and the Research and the Ethics Review Committee of the Ministry of Health and Human Services, Somali Federal Republic (ref. MOH&HS/DGO/1944/Dec/2018). SMART survey participants (all adults) provided verbal informed consent on behalf of their households, in accordance with the SMART survey protocol [8]. Our study relied only on previously collected anonymised and unlinkable datasets, and did not collect any primary data.

## Results

### Crude survey estimates

Table 3 reports descriptive statistics for the 97 eligible mortality surveys. The highest death rates were estimated by surveys done in 2017. Eligible surveys featured a median ratio of U5DR to CDR of around 1.5, with 43% of under 5y deaths among infants; 'injury' caused some 5% of deaths on average. Net migration from households was mostly negative throughout the period, with the exception of 2017. Additional crude trends in demographic indicators over time and by region are shown in Fig C of S1 Text.

### Evolution of population denominators

Between 2013 and 2018, we estimated that Somalia's population increased from 12.0 to 13.5 million, but with a discrepancy among sources of up to 3.5 million (Fig 4). Some regions (Bay, Bakool, Lower Shabelle) saw substantial declines in population, while Banadir and Lower Juba experienced a marked increase in population, reflecting internal displacement during the period (Figs D, F in S1 Text). Overall, 2,319,000 people were reported to have become newly displaced or returned to their district of origin during 2016–2018, with reasons reported as drought (1,262,000, 54.4%), insecurity (698,000, 30.1%), flooding (291,000, 12.6%) and other (68,000, 2.9%). As a proportion of the population, this corresponds to an increase from about 8% to 20% in IDPs and returnees (Fig E in S1 Text), with marked regional differences (Fig F in S1 Text). We estimated that net refugee migration was 144,000 during the analysis period.

### Predictive model

Table 4 summarises final predictive models for both CDR and U5DR (for the latter, the same model specification as for CDR was selected; while other models with equivalent predictive accuracy were available, we considered it advantageous for interpretation that the CDR and U5DR estimates arise from the same statistical process). In both models, death rate increased with incidence of armed conflict, the rate of SAM admissions into nutritional therapy and with non-zero measles incidence, but decreased as the facility-based incidence of malaria increased (see Discussion). Both models featured comparable fit on the training dataset, on 10-fold cross-validation, and on the holdout dataset (Figs G, H in S1 Text).

**Table 3. Crude summary statistics for eligible mortality surveys, overall and by year.**

| Statistic† | Overall | Year | | | | | |
|---|---|---|---|---|---|---|---|
| | | 2013 | 2014 | 2015 | 2016 | 2017 | 2018 |
| Eligible surveys (N) | 97 | 6 | 9 | 8 | 24 | 6 | 44 |
| Crude death rate (per 10,000 person-days) | 0.43 (0.00 to 1.61, 97) | 0.57 (0.11 to 1.61, 6) | 0.46 (0.14 to 0.84, 9) | 0.40 (0.14 to 0.63, 8) | 0.25 (0.00 to 0.6, 24) | 0.59 (0.13 to 0.68, 6) | 0.51 (0.00 to 1.38, 44) |
| Under 5 years death rate (per 10,000 child-days) | 0.66 (0.00 to 2.48, 97) | 0.63 (0.37 to 1.39, 6) | 0.76 (0.32 to 2.00, 9) | 0.45 (0.00 to 1.32, 8) | 0.37 (0.00 to 1.21, 24) | 0.76 (0.33 to 1.44, 6) | 0.72 (0.00 to 2.48, 44) |
| Proportion of under 5y deaths that were among infants <1y | 0.43 (0.00 to 1.00, 59) | no data | no data | no data | 0.50 (0.00 to 1.00, 10) | 0.25 (0.00 to 0.33, 6) | 0.50 (0.00 to 1.00, 43) |
| Household size | 5.6 (4.1 to 6.8, 97) | 5.7 (4.7 to 5.9, 6) | 5.8 (5.0 to 6.1, 9) | 5.9 (5.5 to 6.2, 8) | 5.5 (4.7 to 6.5, 24) | 4.9 (4.5 to 5.6, 6) | 5.6 (4.1 to 6.8, 44) |
| Proportion of children aged under 5y | 0.25 (0.19 to 0.42, 97) | 0.24 (0.22 to 0.26, 6) | 0.26 (0.21 to 0.30, 9) | 0.23 (0.22 to 0.29, 8) | 0.26 (0.20 to 0.34, 24) | 0.29 (0.21 to 0.35, 6) | 0.25 (0.19 to 0.42, 44) |
| Proportion of females in household | 0.51 (0.48 to 0.54, 63) | no data | no data | no data | 0.51 (0.49 to 0.52, 13) | 0.51 (0.51 to 0.52, 6) | 0.51 (0.48 to 0.54, 44) |
| Crude birth rate (per 1000 person-years) | 36.4 (0.0 to 111.6, 97) | 49.1 (26.8 to 78.4, 6) | 39.2 (6.4 to 91.9, 9) | 30.5 (15.0 to 60.8, 8) | 28.2 (0.0 to 111.6, 24) | 37.9 (18.4 to 56.5, 6) | 35.8 (0.0 to 89.9, 44) |
| Net migration rate (per 1000 person-years) | -40.0 (-369.9 to 139.0, 97) | -25.9 (-73.3 to -18.8, 6) | -37.8 (-69.5 to 122.5, 9) | -29.2 (-62.4 to 99.7, 8) | -29.3 (-172.9 to 72.5, 24) | 40.3 (9.8 to 83.1, 6) | -63.3 (-369.9 to 139.0, 44) |
| Injury-specific death rate (per 10,000 person-days) | 0.05 (0.00 to 0.41, 54) | no data | no data | no data | 0.04 (0.00 to 0.27, 12) | 0.03 (0.00 to 0.07, 3) | 0.07 (0.00 to 0.41, 39) |

† Values in cells are median (range, number of surveys with information).

## Estimates of mortality

In the most likely counterfactual scenario, the excess death toll between Jan 2017 and Dec 2018 across Somalia was estimated at 44,700 people, out of some 454,500 total deaths (Table 5). A considerably lower excess was estimated for children under 5y, amounting to about 21% of the all-age excess death toll. However, under a pessimistic scenario the death toll rose to as many as 163,800 (and 61,400 children under 5y). The 2017 excess death toll was about double that in 2018.

When aggregated by region (Table 6), the estimated death rates per 10,000 person-days ivaried from 0.33 in Woqooyi Galbeed to 0.60 in Gedo (CDR), and from 0.42 in Awdal to 1.02 in Hiraan (U5DR), with a markedly lower ratio of U5DR to CDR in the North-East and North-West. The highest excess death rates for all ages were observed in the North-East and in Hiraan, while among children under 5y South-Central Somalia had the highest excess mortality estimates (see also Figs I, J, K, L in S1 Text). Estimates by district are shown in Table A in S1 Text.

During 2014–2018, CDR across Somalia remained within a limited range, but an appreciable elevation (up to 0.09 per 10,000 person-days higher than the most likely counterfactual) was estimated from early 2017 to mid-2018, coinciding with the food security crisis period (Fig 5). The pattern was similar for U5DR, but with a less marked elevation in 2017–2018 (Fig 5).

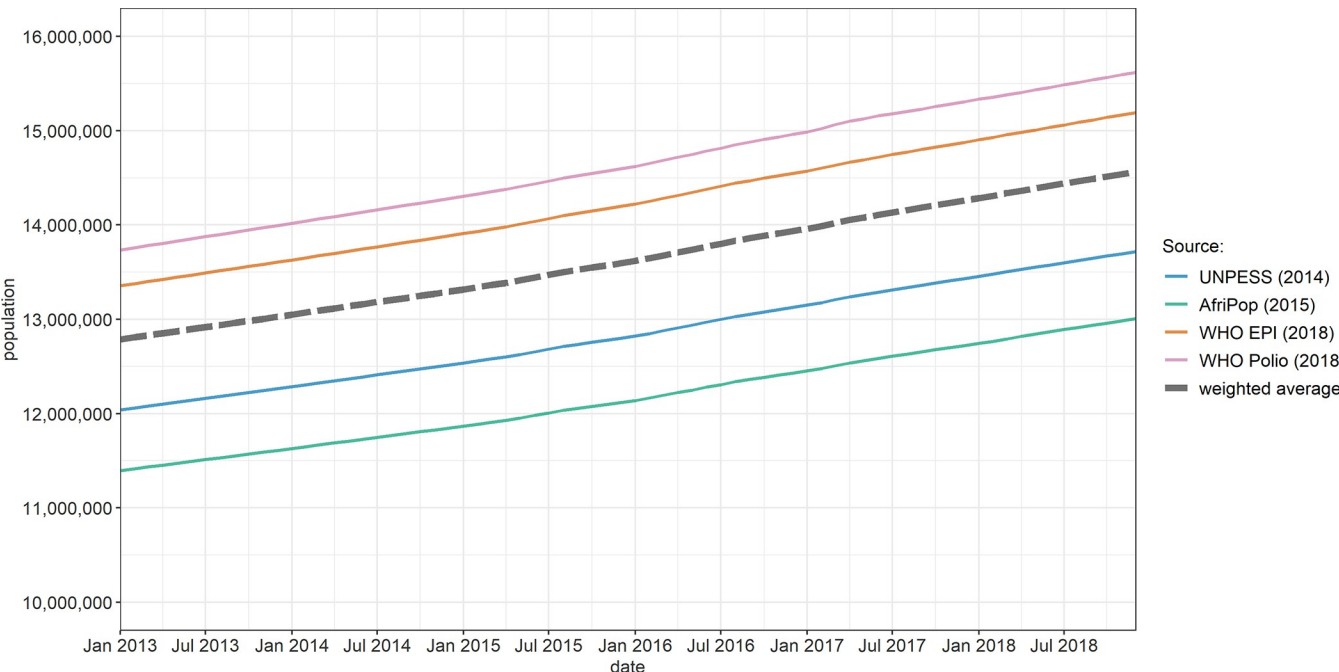

**Fig 4. Evolution of total estimated population, by source.** UNPESS = United Nations Population Estimation Sample Survey. WHO = World Health Organization.

**Table 4.  Predictive models for crude and under 5 years death rate.**

| Predictor | Crude death rate | | | Under 5 years death rate | | |
|---|---|---|---|---|---|---|
| | Rate ratio | 95% CI† | p-value | Rate ratio | 95% CI† | p-value |
| Administrative level | | | | | | |
| Somaliland—Puntland | 1.00 [ref.] | | | 1.00 [ref.] | | |
| South-Central Somalia | 0.97 | 0.77 to 1.24 | 0.833 | 1.54 | 1.08 to 2.20 | 0.016 |
| Incidence of armed conflict events (events per 100,000 person-months) | | | | | | |
| < 0.25 | 1.00 [ref.] | | | 1.00 [ref.] | | |
| 0.25 to 0.49 | 1.00 | 0.81 to 1.23 | 0.985 | 1.26 | 0.94 to 1.69 | 0.119 |
| ≥ 0.50 | 1.60 | 1.30 to 1.98 | <0.001 | 1.44 | 1.08 to 1.92 | 0.013 |
| Rate of admissions of cases of SAM into nutritional therapy (per 100,000 person-months)—lag: 2 months | | | | | | |
| < 100 | 1.00 [ref.] | | | 1.00 [ref.] | | |
| 0.25 to 0.49 | 1.36 | 1.09 to 1.69 | 0.007 | 1.10 | 0.79 to 1.54 | 0.571 |
| ≥ 0.50 | 1.48 | 1.12 to 1.97 | 0.007 | 1.24 | 0.83 to 1.85 | 0.284 |
| Health facility-based incidence rate of malaria (cases per 100,000 person-months) | | | | | | |
| 0 | 1.00 [ref.] | | | 1.00 [ref.] | | |
| 1 to 49 | 0.78 | 0.59 to 1.04 | 0.087 | 0.74 | 0.48 to 1.13 | 0.165 |
| ≥ 50 | 0.71 | 0.54 to 0.94 | 0.017 | 0.55 | 0.37 to 0.83 | 0.004 |
| Incidence rate of measles (reported cases per 100,000 person-months) | | | | | | |
| 0 | 1.00 [ref.] | | | 1.00 [ref.] | | |
| > 0 | 1.27 | 0.99 to 1.62 | 0.056 | 1.37 | 0.94 to 2.00 | 0.100 |

† 95% confidence intervals (based on robust standard errors adjusted for survey intra-cluster correlation).

**Table 5. Total and crisis-attributable deaths, by year and counterfactual scenario.**

| Period | Total deaths (95%CI) | Excess deaths (95%CI), by counterfactual scenario | | |
|---|---|---|---|---|
| | | Most likely | Reasonable worst-case (low counterfactual) | Reasonable best-case (high counterfactual) |
| All ages | | | | |
| 2017 | 226,900 (166,500 to 313,100) | 29,600 (24,500 to 35,600) | 84,700 (69,400 to 101,400) | 16,300 (11,900 to 22,800) |
| 2018 | 227,500 (167,600 to 311,200) | 15,100 (13,100 to 16,500) | 79,100 (66,800 to 90,700) | 3100 (1100 to 5500) |
| Overall | 454,500 (334,100 to 624,400) | 44,700 (37,600 to 52,000) | 163,800 (136,300 to 192,100) | 19,400 (12,900 to 28,300) |
| Children under 5 years | | | | |
| 2017 | 93,800 (65,300 to 136,000) | 6100 (5500 to 6400) | 30,800 (26,600 to 32,400) | 1100 (800 to 1400) |
| 2018 | 92,700 (63,800 to 136,100) | 3200 (2600 to 3900) | 30,600 (26,400 to 31,500) | -1400 (-1600 to -1300) |
| Overall | 186,500 (129,000 to 272,100) | 9300 (8100 to 10,300) | 61,400 (53,100 to 63,900) | -400 (-400 to 100) |

On visual inspection, estimates of the excess death rate correlated broadly with the forecast severity of the crisis as of early 2017, as summarised by the proportion of people projected to be in phases 3 (crisis) or 4 (emergency) of the Integrated Phase Classification (IPC) system for each region of Somalia (Fig 6).

## Discussion

To our knowledge this is the first comprehensive analysis of mortality patterns in Somalia since 2014 and in particular death tolls attributable to the food security crisis of 2017–2018. It follows on from a similar study [1] covering the period October 2010 to April 2012, when a very severe crisis, featuring large pockets of famine, swept through the country [24]. Despite some methodological and data source differences with the 2010–2012 study, the present analysis suggests that the 2017–2018 crisis had a lower mortality impact (about 45,000 excess deaths, compared to 258,000 in 2010–2012, when CDR across South-Central Somalia peaked at some three times the baseline).

Nevertheless, excess mortality estimates for 2017–2018 remain staggering, indicating considerable unmet needs in the response to this crisis. This large death toll, occurring despite an increase in CDR of 'only' about 15%, also reflects a much larger population at risk than in 2010, due to demographic growth and more geographically widespread drought conditions: in 2010–2012, mortality was concentrated in Bay, Lower and Middle Shabelle and Banadir regions, but in 2017–2018 Central and North-East Somalia (Puntland) also seemed heavily affected. During the period 2016–2018 mortality surveillance within IDP camps in the Afgooye corridor reported under 5 deaths rate as several times higher than the emergency threshold on several occasions [25]. The findings of this surveillance system may be an underestimation as deaths in transit were not fully captured.

By contrast to the afore-mentioned 2010–2012 study (1), we found no consistent association of terms of trade (purchasing power) with mortality. The 2010–2012 study suggested that mortality increased steeply below the 10,000 Kcal cereal per daily wage threshold. As shown in Fig M in S1 Text, this threshold was crossed in most regions both in 2008 and in 2010–2011, while in 2017 terms of trade varied little or remained above 20,000 Kcal cereal per daily wage. Despite this, the unusual levels of rainfall, displacement and nutritional therapy admissions in 2017–2018, along with country-wide measles and cholera epidemics (typical of food insecurity crises), all indicate that a substantial crisis did occur in Somalia during this period, with estimated mortality roughly correlating with IPC severity phase classifications.

We conducted a recent ecological analysis exploring drivers of displacement in Somalia between 2016–18. We found that largescale early displacement was strongly associated with a

**Table 6. Crude death rate, excess death rate and excess death toll by region, under the most likely counterfactual scenario, for all ages and children under 5 years.**

| Region | All ages | | | Children under 5 years | | |
|---|---|---|---|---|---|---|
| | Crude death rate† (95%CI) | Excess death rate† (95%CI) | Excess deaths (95%CI) | Under 5y death rate† (95%CI) | Excess under 5y death rate† (95%CI) | Excess deaths under 5y (95%CI) |
| South-Central Somalia | | | | | | |
| Bakool | 0.41 (0.26 to 0.66) | 0.06 (0.05 to 0.06) | 1700 (1400 to 1800) | 0.87 (0.62 to 1.24) | 0.08 (-0.01 to 0.11) | 600 (-100 to 800) |
| Banadir | 0.50 (0.41 to 0.61) | 0.01 (-0.02 to 0.03) | 1200 (-3700 to 4200) | 0.80 (0.61 to 1.07) | 0.02 (0.02 to 0.03) | 900 (800 to 1200) |
| Bay | 0.47 (0.36 to 0.63) | 0.01 (-0.01 to 0.02) | 700 (-400 to 1200) | 0.82 (0.59 to 1.16) | 0.00 (-0.02 to 0.01) | 0 (-400 to 100) |
| Galgaduud | 0.47 (0.34 to 0.68) | 0.04 (0.03 to 0.05) | 1700 (1400 to 1800) | 0.80 (0.55 to 1.17) | 0.03 (-0.01 to 0.05) | 300 (-100 to 500) |
| Gedo | 0.60 (0.46 to 0.79) | 0.06 (0.05 to 0.06) | 2600 (2300 to 2600) | 0.98 (0.70 to 1.38) | 0.10 (0.07 to 0.11) | 1100 (700 to 1100) |
| Hiraan | 0.49 (0.37 to 0.67) | 0.11 (0.09 to 0.14) | 4300 (3700 to 5300) | 1.02 (0.71 to 1.49) | 0.22 (0.18 to 0.26) | 2200 (1700 to 2500) |
| Lower Juba | 0.53 (0.39 to 0.72) | 0.03 (0.02 to 0.03) | 1600 (1000 to 1700) | 0.98 (0.67 to 1.46) | 0.19 (0.16 to 0.23) | 2400 (2000 to 2900) |
| Lower Shabelle | 0.36 (0.25 to 0.53) | 0.03 (0.00 to 0.03) | 2900 (500 to 3500) | 0.85 (0.59 to 1.23) | -0.03 (-0.11 to 0.01) | -900 (-3000 to 200) |
| Middle Juba | 0.39 (0.25 to 0.60) | 0.05 (0.04 to 0.07) | 1500 (1100 to 2000) | 0.92 (0.60 to 1.41) | 0.15 (0.10 to 0.22) | 1100 (700 to 1600) |
| Middle Shabelle | 0.37 (0.27 to 0.52) | 0.04 (0.04 to 0.04) | 1800 (1700 to 1800) | 0.85 (0.59 to 1.23) | 0.01 (-0.05 to 0.04) | 200 (-600 to 500) |
| North-east (Puntland) | | | | | | |
| Bari | 0.45 (0.34 to 0.59) | 0.11 (0.09 to 0.14) | 7300 (5800 to 8900) | 0.50 (0.33 to 0.75) | 0.05 (0.02 to 0.05) | 800 (400 to 900) |
| Mudug | 0.42 (0.28 to 0.64) | 0.12 (0.08 to 0.19) | 7600 (5100 to 11,600) | 0.59 (0.38 to 0.92) | 0.03 (0.00 to 0.09) | 400 (-100 to 1400) |
| Nugaal | 0.50 (0.34 to 0.74) | 0.08 (0.04 to 0.16) | 2300 (1000 to 4200) | 0.65 (0.41 to 1.03) | 0.07 (0.02 to 0.19) | 500 (100 to 1300) |
| North-west (Somaliland) | | | | | | |
| Awdal | 0.35 (0.25 to 0.50) | -0.02 (-0.03 to 0.00) | -1200 (-1800 to -200) | 0.42 (0.26 to 0.68) | -0.03 (-0.04 to 0.00) | -500 (-600 to 0) |
| Sanaag | 0.47 (0.33 to 0.66) | 0.07 (0.04 to 0.13) | 2800 (1500 to 5000) | 0.56 (0.36 to 0.88) | 0.05 (0.01 to 0.12) | 400 (100 to 1200) |
| Sool | 0.49 (0.36 to 0.68) | 0.06 (0.03 to 0.10) | 1600 (800 to 2800) | 0.60 (0.40 to 0.89) | -0.01 (-0.01 to 0.00) | -100 (-100 to 0) |
| Togdheer | 0.45 (0.35 to 0.59) | 0.05 (0.03 to 0.08) | 2500 (1500 to 4100) | 0.53 (0.35 to 0.80) | 0.00 (-0.02 to 0.03) | 0 (-200 to 400) |
| Woqooyi Galbeed | 0.33 (0.24 to 0.45) | 0.02 (0.01 to 0.04) | 1900 (900 to 3800) | 0.46 (0.30 to 0.71) | 0.00 (-0.01 to 0.03) | 0 (-300 to 800) |
| Overall | 0.43 (0.32 to 0.60) | 0.04 (0.04 to 0.05) | 44,700 (37,600 to 52,000) | 0.72 (0.50 to 1.05) | 0.04 (0.03 to 0.04) | 9300 (8100 to 10,300) |

† per 10,000 person-days for CDR, and children under 5y-days for U5DR.

number of crisis related risk factors such as failed rains and subsequent food insecurity while conflict intensity was weakly associated [26]. We plan further analyses to explore reasons the causal relationships among drought, food insecurity and outcomes upstream of mortality, including prevalence of acute malnutrition and epidemic incidence, so as to better understand the dynamics of this latest crisis, and the role of food security versus other drivers. Notably, large-scale displacement in this recent drought has been linked to a 'pull' factor from urban

**all ages**

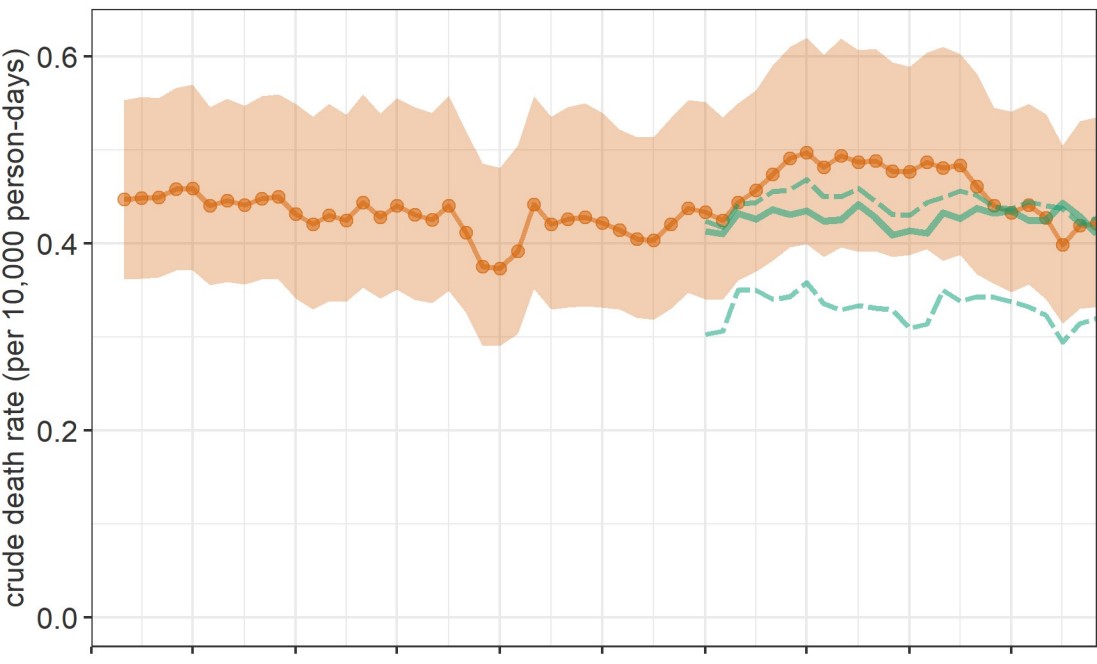

**children under 5y**

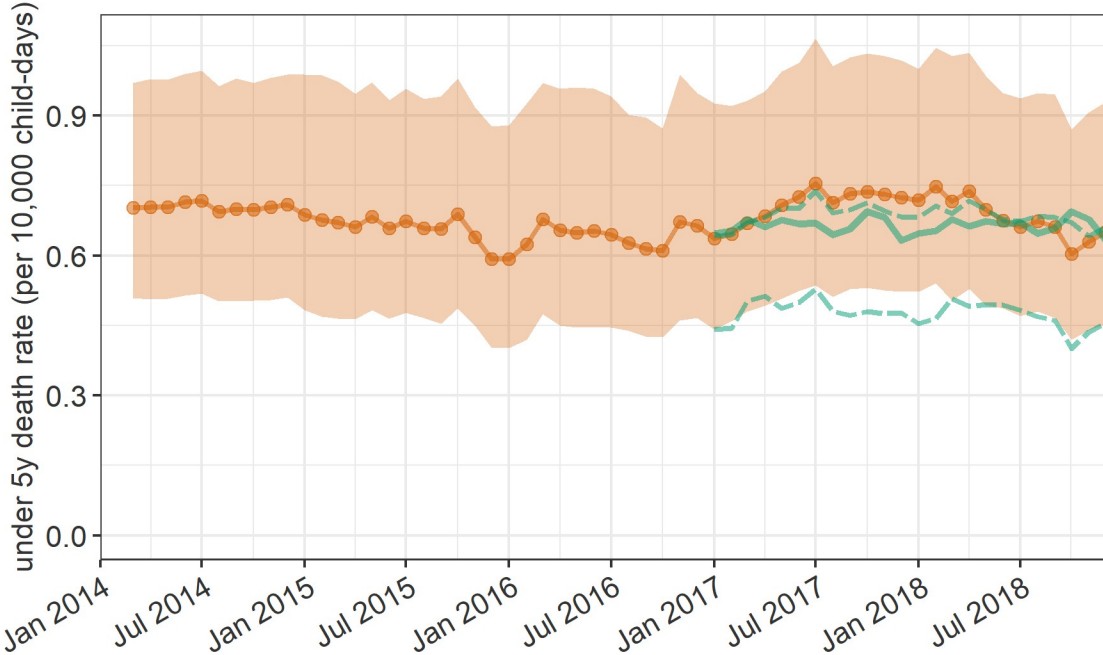

**Fig 5.** Trends in the estimated crude death rate (top panel) and under 5 years death rate (bottom panel). Actual estimates, i.e. under observed conditions, are shown with a dotted solid orange line (point estimate) and orange shaded area (95%confidence interval). The green lines indicate counterfactual levels (most likely scenario: Solid green line; reasonable worst-case and best-case scenario: Dashed green lines).

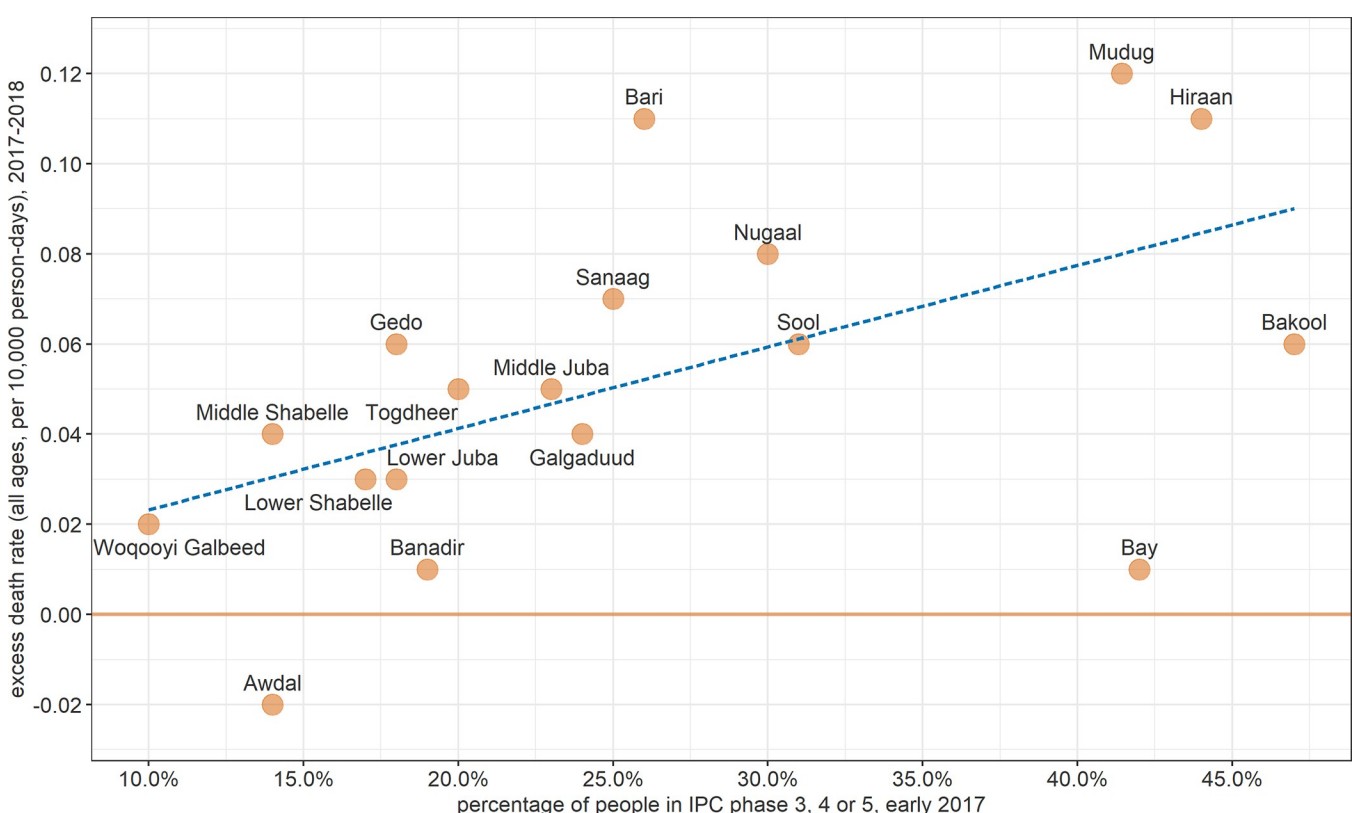

**Fig 6. Correlation between Integrated Phase Classification (IPC) projections (early 2017) and estimated excess death rate among all ages (2017–2018), by region.**

centres offering humanitarian assistance, and the widespread sale of land by impoverished farmers in south-central Somalia [27].

## Study limitations

The validity of the predictive model is central to the estimates' robustness. Cross-validation and the accuracy of prediction on holdout samples support external validity, with little evidence of systematic bias. Internal validity is suggested by the plausible associations in the final models, namely evidence (albeit with weak significance) of higher mortality as a function of armed conflict intensity, admissions for severe acute malnutrition and occurrence of measles. Less interpretably, mortality was associated with increasing malaria incidence in health facilities: this may be a proxy for levels of drought (lower rainfall could have led to decreased mosquito breeding and thus malaria transmission: otherwise put, malaria transmission may have correlated with better-than-average food security) or better access to health services if malaria caseload was a proxy for outpatient service utilisation.

A possible source of bias (albeit with unclear directionality) is inaccuracy in the source demographic estimates. We attempted to mitigate this by averaging the four available estimates based on their assumed robustness. However, the observed discrepancy among estimates suggests considerable uncertainty, particularly when considering estimates at the regional and district scale. Moreover, our redistribution of population among districts based on internal displacement movements rests on the accuracy of PRMN reports, which are not based on statistically representative estimation methods, but rather ground informants. Though the

Table 7. Assessment of systematic bias in the estimates.

| Direction of bias | Details |
|---|---|
| Known or suspected reasons for underestimation | Under-reporting of child deaths, particularly among neonates and infants<br>Selection bias due to exclusion of insecure or inaccessible areas of a district from the survey's sampling frame (information on this was very scant, as survey reports were mostly not available to us)<br>Exclusion of refugees in Somalia and Somali refugees abroad from the analysis |
| Known or suspected reasons for overestimation | [none] |
| Other possible biases with unclear directionality | Inaccuracy in demographic estimates<br>Reduction in birth rate and increase in mortality during the crisis period, leading to lower rate of population growth and thus overestimation of population<br>Inaccuracy in reported internal displacement figures, particularly for returnees to districts of origin<br>Faulty assumptions about the counterfactual baseline |
| Likely overall extent and direction of bias | Mild-moderate underestimation in the actual and counterfactual mortality estimates; see sensitivity analysis (Figs N, O in S1 Text) for overall effect on excess mortality estimates |

PRMN project captures both departures and returns, it is plausible that the latter flows would be less systematically reported.

We estimated a relatively low excess mortality among children under 5y, compared to among all ages: by comparison, about half of excess deaths in 2010–2012 were in this age group, and meta-analyses of SMART surveys have generally found the U5DR to CDR ratio to be around two, rather than 1.5 in this study [28,29]. A similar analysis in South Sudan [23] suggested that deaths among infants had been substantially underestimated by SMART surveys, possibly due to stigma associated with child deaths or faulty administration of the mortality questionnaire. It is possible that surveys included in this analysis were also subject to some underestimation in neonatal and infant events, as suggested by a lower crude birth rate (36 versus 43 per 1000 person-years) and proportion of infants among under 5y deaths (43% versus 60%) in comparison to estimates by demographic models for Somalia [21].

To explore sensitivity of our estimates to key potential error sources, we (i) assumed different levels of bias in source demographic estimates and reported displacement movements, and (ii) varying proportions of under-reporting among deaths under 5y (see Checchi et al. [7] for detail). Generally, excess mortality estimates appeared relatively insensitive to bias in displacement figures, but decreased substantially as bias in demographic estimates (either over- or underestimation) increased (Fig N in S1 Text). As under-estimation in U5DR rose from 0% to 50%, a doubling in the excess death toll under 5y was projected, but all-age death tolls increased only moderately (Fig O in S1 Text). Our overall assessment of systematic bias in our estimates is presented in Table 7.

## Conclusions

This study finds evidence of elevated mortality during a drought-triggered crisis in Somalia over 2017–2018, despite a lesser effect on food security than in previous similar events, and a more proactive and far-reaching humanitarian response [30]. Our findings indicate that even at moderate levels of population stress (for example, IPC phase 3), excess mortality accrues. Therefore, even if the sole aim of humanitarian actors is to support survival, a response needs to be implemented to scale very early in the curve of deterioration in food security and other upstream crisis indicators [31]. While our analysis does not have granularity below district

level, we speculate that mortality risk is likely to be clustered within particularly vulnerable communities and households: more specific targeting of limited resources to support these communities and individuals would probably increase the cost-effectiveness and efficiency of humanitarian responses, particularly in food insecurity crises.

Somalia and other fragile regions of the world appear to face an increased threat of drought due to climate change, compounded by global inflation in food and commodity prices. At the time of writing, Somalia and parts of the Horn of Africa face a new, potentially severe food insecurity crisis [32]. Understanding the mortality impact of these events can support rational resource allocation and benchmark the adequacy of humanitarian responses. As such, mortality analyses should arguably become a systematic component of monitoring and evaluation in drought-triggered crises. It is also critical that the narrative of Somalia's crisis not lose sight of human security as a key driver of mortality, as shown in our model. Armed conflict and the securitisation of much of Somalia exacerbate the severity of drought and underlie the chronic vulnerability of Somali people to unfavourable climate: the restoration of peace and governance to all of Somalia holds perhaps the greatest potential for attenuating the impact of droughts and other natural disasters on its population.

## Supporting information

**S1 Text. Additional methods and results.**
(DOCX)

## Acknowledgments

We are grateful to Anna Carnegie for project management support and the Federal Ministry of Health and Social Services of Somalia for facilitating data acquisition. We are also grateful to Amy Gimma and Christopher Jarvis for technical advice. Lastly, we thank all agencies that provided data, and the field data collectors from the Food Security and Nutrition Analysis Unit and other organisations who collected primary mortality household data.

The authors are solely responsible for the analyses presented here, and acknowledgment of data sources does not imply that the agencies providing data endorse the results of the analysis. The opinions expressed in this article are those of the authors only and do not necessarily represent the decisions, policies, or views of the London School of Hygiene & Tropical Medicine.

## Author Contributions

**Conceptualization:** Abdihamid Warsame, Francesco Checchi.

**Data curation:** Abdihamid Warsame, Séverine Frison, Francesco Checchi.

**Formal analysis:** Séverine Frison, Francesco Checchi.

**Funding acquisition:** Francesco Checchi.

**Investigation:** Francesco Checchi.

**Methodology:** Francesco Checchi.

**Project administration:** Abdihamid Warsame, Francesco Checchi.

**Writing – original draft:** Abdihamid Warsame, Francesco Checchi.

**Writing – review & editing:** Abdihamid Warsame, Séverine Frison, Francesco Checchi.

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
