## [Decision Letter · Decision Letter 0]

1 Nov 2022

PGPH-D-22-01475

Drought, armed conflict and population mortality in Somalia, 2014-2018: a statistical analysis

Dear Dr. Checchi

Thank you for submitting your manuscript to PLOS Global Public Health. After careful consideration, we feel that it has merit but does not fully meet PLOS Global Public Health’s publication criteria as it currently stands. Therefore, we invite you to submit a revised version of the manuscript that addresses the points raised during the review process.

We look forward to receiving your revised manuscript.

Kind regards,

Andreas K Demetriades, MBBChir, MPhil, FRCSEd, FEBNS.

Academic Editor

Journal Requirements:

a) State the initials, alongside each funding source, of each author to receive each grant. For example: "This work was supported by the National Institutes of Health (####### to AM; ###### to CJ) and the National Science Foundation (###### to AM)."

2. Please update your online Competing Interests statement. If you have no competing interests to declare, please state: “The authors have declared that no competing interests exist.”

3. Please provide separate figure files in .tif or .eps format only and ensure that all files are under our size limit of 10MB.

4. We noticed that you used “unpublished” / “unpublished data” in the manuscript. We do not allow these references, as the PLOS data access policy requires that all data be either published with the manuscript or made available in a publicly accessible database. Please amend the supplementary material to include the referenced data or remove the references.

5. Some material included in your submission may be copyrighted. According to PLOS’s copyright policy, authors who use figures or other material (e.g., graphics, clipart, maps) from another author or copyright holder must demonstrate or obtain permission to publish this material under the Creative Commons Attribution 4.0 International (CC BY 4.0) License used by PLOS journals. Please closely review the details of PLOS’s copyright requirements here: PLOS Licenses and Copyright. If you need to request permissions from a copyright holder, you may use PLOS's Copyright Content Permission form.

Potential Copyright Issues:

Fig S4 and Fig S5: please (a) provide a direct link to the base layer of the map (i.e., the country or region border shape) and ensure this is also included in the figure legend; and (b) provide a link to the terms of use / license information for the base layer image or shapefile. We cannot publish proprietary or copyrighted maps (e.g. Google Maps, Mapquest) and the terms of use for your map base layer must be compatible with our CC-BY 4.0 license. 

Additional Editor Comments (if provided):

as per peer review

Reviewers' comments:

Reviewer's Responses to Questions

**Comments to the Author**

1. Does this manuscript meet PLOS Global Public Health’s publication criteria? Is the manuscript technically sound, and do the data support the conclusions? The manuscript must describe methodologically and ethically rigorous research with conclusions that are appropriately drawn based on the data presented.

Reviewer #1: Yes

2. Has the statistical analysis been performed appropriately and rigorously?

Reviewer #1: Yes

3. Have the authors made all data underlying the findings in their manuscript fully available (please refer to the Data Availability Statement at the start of the manuscript PDF file)?

Reviewer #1: Yes

4. Is the manuscript presented in an intelligible fashion and written in standard English?

Reviewer #1: Yes

5. Review Comments to the Author

Reviewer #1: Thank you for the opportunity to review this manuscript on estimating the excess mortality in Somalia attributable to drought-triggered crisis in 2017-2018. This study used data sources from various agencies to tackle this challenging topic, adopting a robust methodological approach to answering the research question. It is an important study to understand the complex situation that occurred in Somalia. While the general narrative of the manuscript is good, there are some areas that I have think can improve the readers' understanding of the study.

Under 'study design', I suggest adding in the github repository (https://github.com/francescochecchi/mortality_small_area_estimation) rather than only referencing the index paper for greater transparency.

I think Figure S1 is an important figure because it shows the distribution of data available per district, which aid interpretation of estimates described later in this manuscript. Please put this figure as a main figure in the manuscript. Figure S1 should have a scale for the colour. I believe the right Y-axis is the region in Somalia - please add the axis label.

Under 'ground mortality data', the authors state that there were 104 surveys excluded. However, the numbers of subsequent surveys excluded do not add up (i.e. 87 + 6). Please double check the numbers and provide an accurate description of the data sources. Perhaps just adding in the 11 from Figure 3 would do.

Main shortcoming is that 104 out of 201 surveys were excluded and 87 becuase they were stratified by livelihood rather than district as predictor / population datasets not available for livelihood.

Please consider combining figures 2 & 3 to make space for Figure S1.

'We used manual imputation and moderate spline smoothing to remove missingness and minimise outliers'. - I don't understand what this means. Please describe the imputation procedure in more detail and how splines were used to minimise outliers e.g. how did you define an appropriate spline function.

Predictive models - please provide details about how the model selection process took place. For example, it would be helpful to understand what metrics were used to decide on predictive accuracy. Also, please specify what comparable predictive accuracy means in the context of choosing a model. If possible, specifying the range of 'accuracies' within the top 20% best-fitting alternatives would be helpful to the readers. The calibration plots are informative but I cannot understand how you compared different predictive models.

There are many reasons against a 80/20 splitting for internal validation. Given you have performed 10-fold cross validation, I wonder if you can simply remove the the 80/20 split. To confirm, were the models developed on the 80% data available?

In the sensitivity analyses, please describe what level of augmentation carried out. It is difficult to understand how this process happened with the current description.

Given the datasets are either in the public domain or shared in fully anonymised format, could the analytic script be shared on github or similar platform for greatest transparency?

Under 'predictive model' section, I do not understand how Table 4 is helpful. The rationale for these models are for estimating mortality. I would find presenting the poisson regression formulae more useful, but probably as supplement.

Again, I think S4 & S5 are very helpful figures. Could these be combined into one main figure? Ideally there would be a figure for U5DR as well.

Figure 5 needs some figure legends and captions.

In the discussion, there is a section discussing the finding on the association between trade and mortality. This is not described in the body of the manuscript. The reference to 'the 2010-2012' study is confusing as there is no citation provided. If this is deemed important in the context of the paper, please separately report this finding in the results section, as well as describing the method to draw on your results. Otherwise, please consider removing this from the discussion.

Figure 6, i.e. new findings, should be presented in the results section and not in the discussion. The correlation described should be specified in the methods section - how was this done and what metric did you use. This is a very important figure because much of your conclusion is drawn from this figure. So clear explanation on the methods and the results would greatly improve the interpretation of this figure.

I think Table 7 can be summarised into a paragraph, rendering Table 7 not necessary.

Throughout the text, please cite the specific figure/table in the supplement rather than citing S1 Appendix. There are also some abbreviations in the figures, so please provide figure legends that specify these abbreviations.

6. PLOS authors have the option to publish the peer review history of their article (what does this mean?). If published, this will include your full peer review and any attached files.

**Do you want your identity to be public for this peer review?** For information about this choice, including consent withdrawal, please see our Privacy Policy.

Reviewer #1: **Yes: **Michael Tin Chung Poon

---

## [Decision Letter · Decision Letter 1]

13 Mar 2023

PGPH-D-22-01475R1

Drought, armed conflict and population mortality in Somalia, 2014-2018: a statistical analysis

Dear authors,

Thank you for submitting your manuscript to PLOS Global Public Health. After careful consideration, we feel that it has merit but does not fully meet PLOS Global Public Health’s publication criteria as it currently stands. Therefore, we invite you to submit a revised version of the manuscript that addresses the points raised during the review process.

We look forward to receiving your revised manuscript.

Kind regards,

Andreas K Demetriades, MBBChir, MPhil, FRCSEd, FEBNS.

Academic Editor

Journal Requirements:

1. Our staff editors have determined that your manuscript is likely within the scope of our Climate Change and Human Health Call for Papers. This editorial initiative is headed by a team of Guest Editors for PLOS GPH: Renzo Guito (St. Luke's Medical Center College of Medicine, Philippines) and Tolu Oni (University of Cambridge) as well as the Guest Editor for PLOS Climate Anna Stewart Ibarra (Inter-American Institute for Global Change Research). The Collection will feature research that addresses all aspects of the intersection between climate and health, from the changing burden of communicable and non-communicable disease to the impacts of extreme events on health systems, as well as research that assesses potential adaptations to build healthier and more resilient societies. Additional information can be found on our announcement page: https://collections.plos.org/call-for-papers/climate-change-and-human-health/. 

If you would like your manuscript to be considered for this collection, please let us know in your cover letter and we will ensure that your paper is treated as if you were responding to this call.  Please note that being considered for the Collection does not require additional peer review beyond the journal’s standard process and will not delay the publication of your manuscript if it is accepted by PLOS GPH. If you would prefer to remove your manuscript from collection consideration, please specify this in the cover letter.

Additional Editor Comments (if provided):

Reviewers' comments:

Reviewer's Responses to Questions

**Comments to the Author**

1. If the authors have adequately addressed your comments raised in a previous round of review and you feel that this manuscript is now acceptable for publication, you may indicate that here to bypass the “Comments to the Author” section, enter your conflict of interest statement in the “Confidential to Editor” section, and submit your "Accept" recommendation.

Reviewer #2: All comments have been addressed

2. Does this manuscript meet PLOS Global Public Health’s publication criteria? Is the manuscript technically sound, and do the data support the conclusions? The manuscript must describe methodologically and ethically rigorous research with conclusions that are appropriately drawn based on the data presented.

Reviewer #2: Yes

3. Has the statistical analysis been performed appropriately and rigorously?

Reviewer #2: Yes

4. Have the authors made all data underlying the findings in their manuscript fully available (please refer to the Data Availability Statement at the start of the manuscript PDF file)?

Reviewer #2: Yes

5. Is the manuscript presented in an intelligible fashion and written in standard English?

Reviewer #2: Yes

6. Review Comments to the Author

Reviewer #2: The authors propose a statistical analysis of the mortality and excess mortality in Somalia, durring the drough, malaria and cholera epidemics, and peristent armed conflict related crisis, which had taken place from 2014-2018.

The authors also developed a prediction model to estimate crude and under 5 years death rate and overall death tolls, and also noted a significant decrease of excess mortality in the study period when compared to the previously studied 2010-2012 crisis.

The study is exceptionally well carried, and the authors resolved comments from the previous review round adequatelly. I believe that the Figures 2 and 3 could not be merged by any means, thus leaving no space to move Figure S1 to the main manuscript.

Please provide the Supplementary map Figure on the U5DR (although similar to the CDD) to retain equal attention to both estimates.

7. PLOS authors have the option to publish the peer review history of their article (what does this mean?). If published, this will include your full peer review and any attached files.

**Do you want your identity to be public for this peer review?** For information about this choice, including consent withdrawal, please see our Privacy Policy.

Reviewer #2: No

---

## [Decision Letter · Decision Letter 2]

23 Mar 2023

Drought, armed conflict and population mortality in Somalia, 2014-2018: a statistical analysis

PGPH-D-22-01475R2

Dear authors

We are pleased to inform you that your manuscript 'Drought, armed conflict and population mortality in Somalia, 2014-2018: a statistical analysis' has been provisionally accepted for publication in PLOS Global Public Health.

Best regards,

Andreas K Demetriades, MBBChir, MPhil, FRCSEd, FEBNS.

Academic Editor

Thanks for addressing the suggested changes.

This is recommended for publication.

Reviewer Comments (if any, and for reference):

Reviewer's Responses to Questions

**Comments to the Author**

1. If the authors have adequately addressed your comments raised in a previous round of review and you feel that this manuscript is now acceptable for publication, you may indicate that here to bypass the “Comments to the Author” section, enter your conflict of interest statement in the “Confidential to Editor” section, and submit your "Accept" recommendation.

Reviewer #2: All comments have been addressed

2. Does this manuscript meet PLOS Global Public Health’s publication criteria? Is the manuscript technically sound, and do the data support the conclusions? The manuscript must describe methodologically and ethically rigorous research with conclusions that are appropriately drawn based on the data presented.

Reviewer #2: Yes

3. Has the statistical analysis been performed appropriately and rigorously?

Reviewer #2: I don't know

4. Have the authors made all data underlying the findings in their manuscript fully available (please refer to the Data Availability Statement at the start of the manuscript PDF file)?

Reviewer #2: Yes

5. Is the manuscript presented in an intelligible fashion and written in standard English?

Reviewer #2: Yes

6. Review Comments to the Author

Reviewer #2: The authors implemented the Figure, and also resolved previously raised concerns.

7. PLOS authors have the option to publish the peer review history of their article (what does this mean?). If published, this will include your full peer review and any attached files.

**Do you want your identity to be public for this peer review?** For information about this choice, including consent withdrawal, please see our Privacy Policy.

Reviewer #2: No
